# The introduction of the Barbier reaction into polymer chemistry

Xiao-Li Sun[1], Dong-Ming Liu[1], Di Tian[1], Xiao-Yun Zhang[1], Wei Wu[1] & Wen-Ming Wan [1]

The Barbier reaction, a widely utilized reaction for carbon–carbon bond formation, has played important roles in modern organic chemistry for more than a century. Here, we show its successful introduction to polymer chemistry. Through one-pot Barbier polyaddition (both $A_2 + B_2$ type and AB type) of monomers containing an organic halide and a benzoyl group, a series of phenylmethanol group containing polymers, including polymonophenylmethanol (PMPM), polydiphenylmethanol (PDPM), and polytriphenylmethanol (PTPM), have been synthesized. Para-PTPM exhibits interesting aggregation-induced emission, tunable thermo-responsive over a wide temperature range, sensory, luminescence enhancement of fluorescent dye in solid-state and processing properties. This significantly expands the libraries of monomer and polymer, and opens up an avenue for the design and application of functional polymer materials.

[1] State Key Laboratory of Heavy Oil Processing, Centre for Bioengineering and Biotechnology, and College of Science, China University of Petroleum (East China), 66 Changjiang West Road, Qingdao Economic Development Zone, Qingdao, Shandong 266580, People's Republic of China. Correspondence and requests for materials should be addressed to W.-M.W. (email: wanwenming@upc.edu.cn)

Classic organic reactions for carbon–carbon bond forma-tion, including atom transfer radical addition[1, 2], radical addition-fragmentation[3, 4], olefin metathesis[5–7], Suzuki coupling[8–13], Stille coupling[14–17] etc., have been extensively uti-lized in polymer synthesis. Introducing small molecular carbon–carbon bond formation reactions to polymer synthesis has become a desirable goal for polymer chemists because it greatly expands the structure and property library of polymeric materials. The Barbier reaction[18–22], an organic reaction for carbon–carbon bond formation through metal-mediated addition of an organic halide to a carbonyl compound as an electrophilic substrate, has been widely used in organic chemistry since 1899. In comparison with the analogous Grignard reaction, the Barbier reaction benefits from tolerance to functional groups and moisture, widely applicable metals and one-pot preparation. Even though the Barbier reaction has played an important role in the field of organic chemistry for more than a century, its utilization in polymer synthesis has to our knowledge yet to be explored. Considering the versatility of the Barbier reaction in modern organic chemistry, its introduction into polymer science therefore will significantly expand the libraries of both monomer and polymer, and open up an avenue in the design and application of functional polymer materials. Here, we report the introduction of the Barbier reaction to polymer chemistry. Through the Barbier polyaddition (both $A_2+B_2$ type and AB type) of monomers containing both an organic halide and a benzoyl group, a series of phenylmethanol polymers have been successfully prepared, which show potential to be a special type of stimuli-responsive polymer in the field of dual thermo-responsive materials (both lumines-cence and turbidity).

## Results

**Introduction of Barbier reaction to polymer synthesis.** As a versatile carbon–carbon bond formation reaction, the Barbier reaction has been widely used in two-component addition reac-tion and intramolecular cyclization reactions for more than a century, as shown in Fig. 1 (previous work). However, it has not been used in polymer synthesis. We assume that the alternative pathway of intramolecular cyclization reaction is the major obstacle that obstructs the continuous addition, which then results in oligomers with limited molecular weight. To overcome the potential intramolecular cyclization, a series of monomers have been selected with halide and carbonyl groups in the para-positions (Table 1 entry 1, 2, and 4) or meta-positions (Table 1 entry 3) of one rigid benzene ring, or with the benzene rings as steric hindrance groups (Table 1 entry 5–8). The steric hindrance effect of phenyl groups either in the main chain or as side groups of the resulting polymers would favor polymerization rather than cyclization in the presence of rigid benzene rings. All the monomers used in this work were selected so that they could be based on commercially available compounds, from which it should be clear that there is a widespread and accessible chem-istry to allow the Barbier reaction to contribute to the design and application of polymers.

Barbier reactions of different monomers were carried out in refluxing THF at 80 °C for 24 h and a series of phenylmethanol polymers, including polymonophenylmethanol (PMPM), poly-diphenylmethanol (PDPM), and polytriphenylmethanol (PTPM), has been prepared. Taking the Barbier polymerization of 4-bromobenzophenone monomer in the synthesis of PTPM-1 polymer as an example, the successful polymer synthesis is verified by the disappearance of sharp ¹H NMR signals of monomer (Fig. 2a) and the appearance of broader aromatic signals at ~7.5–7.0 ppm and hydroxide proton at ~3.0 ppm (Fig. 2b). The successful formation of polymer is further confirmed by GPC characterization with a $M_n$ of 4700 and PDI of 1.28 (Fig. 2c). For a better understanding of Barbier reaction in polymerization, the polymerization mechanism is illustrated in Fig. 1 (this work), via both AB type and $A_2+B_2$ type Barbier

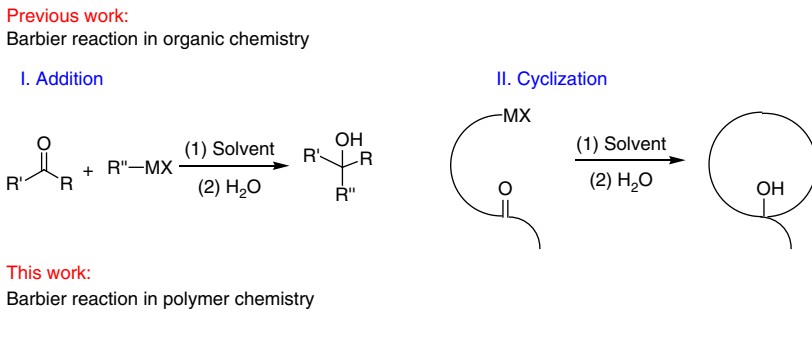

**Previous work:**
Barbier reaction in organic chemistry

**Fig. 1** Introducing the Barbier reaction to polymer science. Widespread Barbier reaction in organic chemistry with small molecules synthesized in literatures; successful Barbier reaction in polymer chemistry (via both AB type and $A_2+B_2$ type Barbier polyaddition) with polymers synthesized in this work

**Table 1 The monomer structures and results of polymers synthesized via Barbier polyaddition in THF at 80 °C for 24 h**

| Entry | Monomer | Polymer | Abbreviation[a] | Yield(%) | $M_{n, NMR}$(Da)[b] | $M_n$(Da)[c] | PDI[c] | FL |
|---|---|---|---|---|---|---|---|---|
| 1 | | | PTPM-1 | 86.4 | 5600 | 5300 | 1.28 | AIE |
| 2 | | | PTPM-1' | 83.6 | 4600 | 4600 | 1.31 | AIE |
| 3 | | | PTPM-2 | 76.6 | 5300 | 4800 | 1.25 | ACQ |
| 4 | | | PDPM | 61.8 | 5600 | 4300 | 1.21 | Weak |
| 5 | | | PMPM | 67.7 | 5400 | 5500 | 1.39 | Weak |
| 6 | | | PXPM-1 | 55.7 | 6500 | 5800 | 1.47 | Weak |
| 7 | | | PXPM-2 | 57.4 | 5600 | 4900 | 1.40 | Weak |
| 8 | | | PXPM-3 | 57.0 | 5200 | 5600 | 1.45 | Weak |

[a]Triphenylmethanol polymer (PTPM), diphenylmethanol polymer (PDPM), monophenylmethanol polymer (PMPM), and xylylene-*alt*-phenylmethanol polymer (PXPM)
[b]Calculated from [1]H NMR spectroscopy by comparing the integral ratio between terminal group and polymer
[c]Measured by GPC

polyaddition between Grignard reagent and carbonyl group in one pot. As the Grignard reagent forms, the addition of highly reactive Grignard reagent to the carbonyl group takes place successively in one pot. Because of the bifunctionality of the monomers, either AB type or $A_2+B_2$ type, continuous addition reactions between Grignard reagents and carbonyl groups happen, i.e., polyaddition mechanism, resulting in successful polymer synthesis. To explain the obtained polymer structure and further confirm the proposed polymerization mechanism, FT-IR and XPS O1s characterizations were carried out. From the terminal benzophenone group existence (~7.8 ppm in Fig. 2b), the carbonyl group existence (verified by FT-IR and XPS O1s spectra in Fig. 2d, e, respectively), and the appearance of C–OH group (verified by [1]H NMR, FT-IR, and XPS O1s spectra in Fig. 2b, d, e, respectively), the polymer structure is verified to contain C–OH group and terminal benzophenone group, and the polymerization mechanism is verified as Barbier polyaddition. The relatively high ratio of C=O peak/C–O peak in XPS O1s spectra should be due to the uneven distribution of terminal C=O group and repeating C–O groups in the polymer materials obtained through precipitation method. Other examples of chemical structures of monomers and the corresponding polymers synthesized via Barbier polyaddition are listed in Table 1. From the comparison of [1]H NMR spectra of these monomers and corresponding products (Supplementary Figs. 1–8), the products clearly exhibit much broader signals for aromatic protons at ~8.0–6.5 ppm, hydroxide proton at ~3.5–2.5 ppm and alkyl protons at ~2.5–1.0 ppm, which is typical evidence of successful polymer synthesis. For further confirmation of the successful synthesis of polymers via Barbier polyaddition, GPC curves are also shown in Supplementary Figs. 1–8 and Table 1. From the monomer structures, it is obvious that Barbier polyaddition works for different halides (both chloride and bromide, Table 1 entry 1 and 2), different substituents (Table 1), and different monomer types (both AB type and $A_2+B_2$ type, Table 1). The further versatility of Barbier polyaddition is that a series of hydroxide group containing polymers can be easily prepared with different substituents, resulting in easy molecular design and functional application. It is worth mentioning that molecular weight distributions of the synthesized polymers are relatively narrow (PDI < 1.5) (Supplementary Fig. 9). Accurate molecular weight of PTPM-1 was further verified to be 9900 Da by static light scattering (Supplementary Fig. 10).

**Aggregation-induced emission (AIE) property of para-triphenylmethanol polymer.** By introducing the Barbier reaction to polymer chemistry, a series of phenylmethanol polymers has been prepared. An intriguing aspect of these polymers is seen in their photophysical properties (Supplementary Table 1 and Supplementary Figs. 11–17). Among these phenylmethanol polymers, PMPM, PDPMs, and alternating copolymers (PXPMs) are weakly luminescent (Supplementary Table 1 entry 6–10 and Supplementary Figs. 13–17). The para-PTPM (PTPM-1) exhibits AIE behavior (Table 1 entry 1 and 2 and Supplementary Fig. 11), while meta-PTPM (PTPM-2) exhibits aggregation-caused quenching (ACQ) behavior (Table 1 entry 3 and Supplementary Fig. 12). Obviously, the photophysical properties of these phenylmethanol polymers benefit from the increase of the amounts of phenyl groups on carbon center of methanol. Considering the chemical similarity of triphenylmethanol moieties of

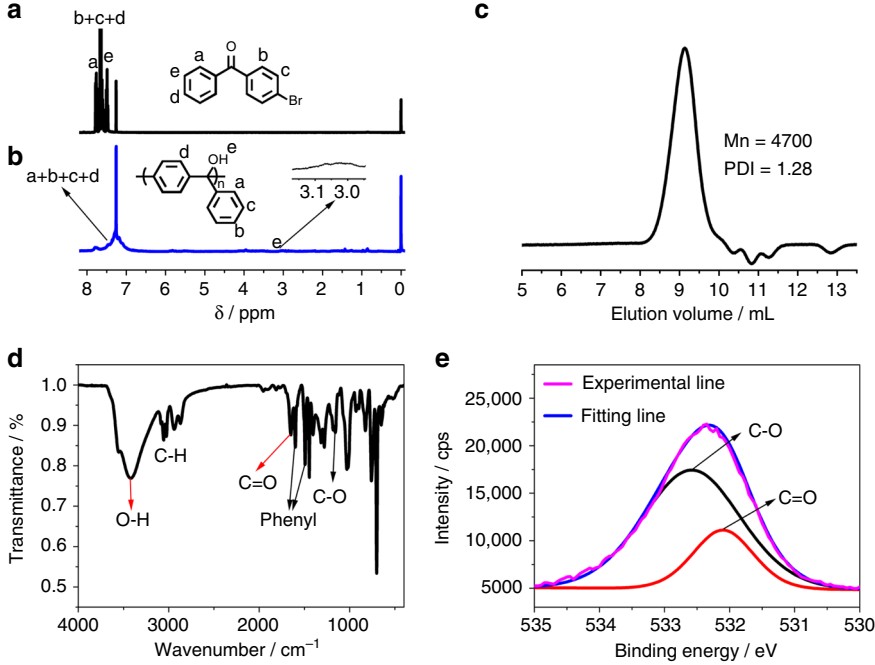

**Fig. 2** Characterizations of PTPM-1. $^1$H NMR spectra of 4-bromobenzophenone (**a**) and PTPM-1 (**b**) in CDCl$_3$, GPC curve of PTPM-1 (**c**) in THF, FT-IR spectrum of PTPM-1 (**d**) and XPS O1s spectra of PTPM-1 (**e**)

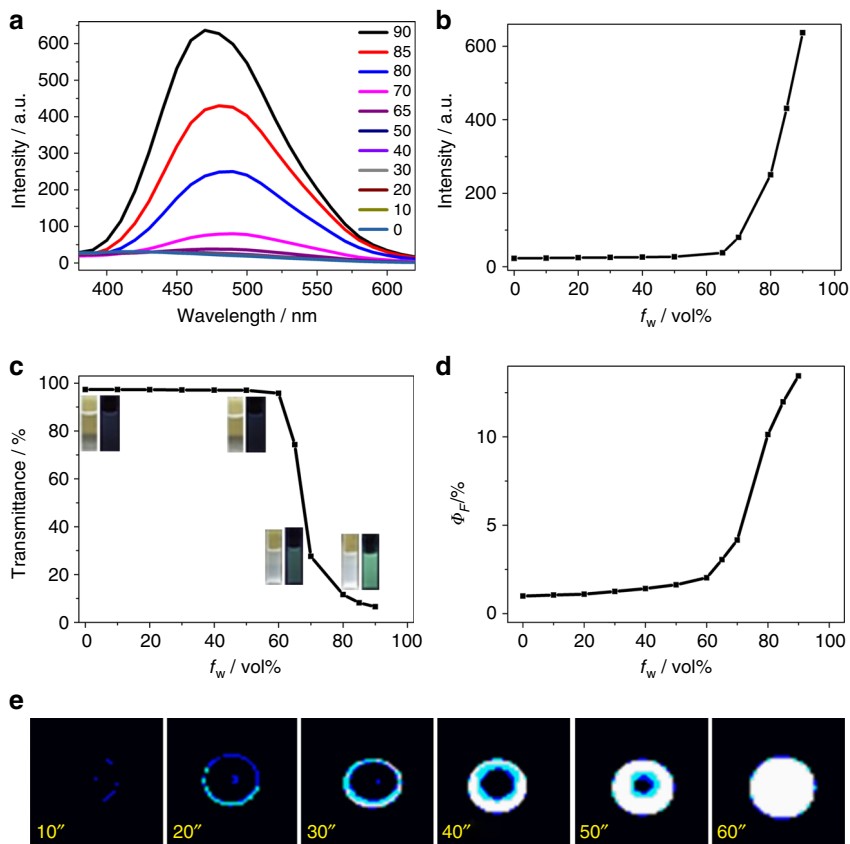

**Fig. 3** AIE properties of PTPM-1. **a** Emission spectra (excited @346 nm), **b** emission intensities (excited @346 nm), **c** transmittance @500 nm wavelength and digital photos (under sunlight and irradiation with UV lamp @365 nm), and **d** fluorescent quantum yields ($\Phi_F$) of PTPM-1 (0.1 mg/mL) in water/THF mixtures with different water volume fractions ($f_W$); **e** digital photos of one drop of PTPM-1 solution (10 mg/mL in THF) on thin-layer chromatography plate with different evaporation timescale at room temperature (under irradiation with UV lamp @365 nm)

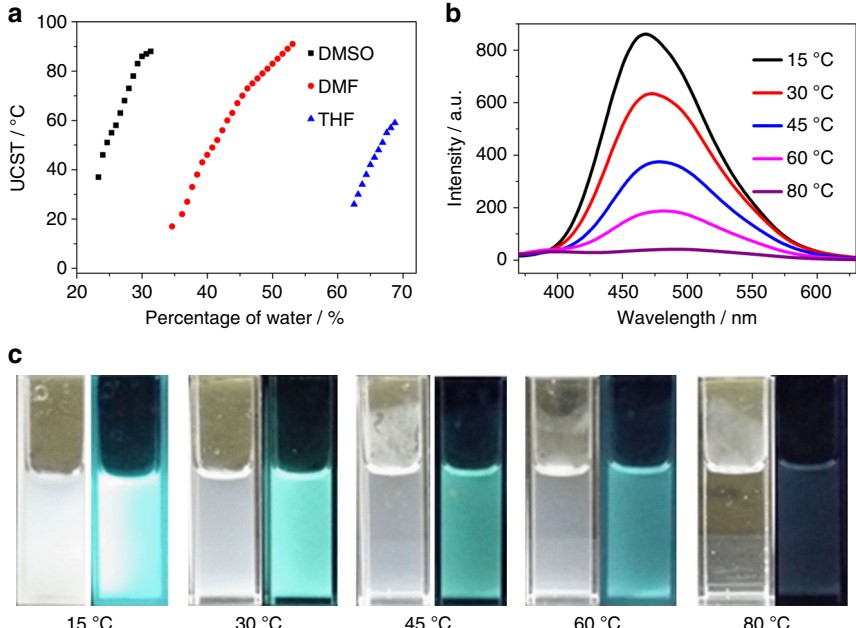

**Fig. 4** Dual (fluorescent and thermo-) responsive properties of PTPM-1. **a** Tunable UCST values of PTPM-1 (2 mg/mL) in DMSO, DMF, and THF with different amounts of water (vol%); **b** emission spectra (excited @346 nm) and **c** digital photos (under sunlight and irradiation with UV lamp @365 nm, respectively) of PTPM-1 (2 mg/mL) in DMF with 45% of water (vol%) at different temperatures

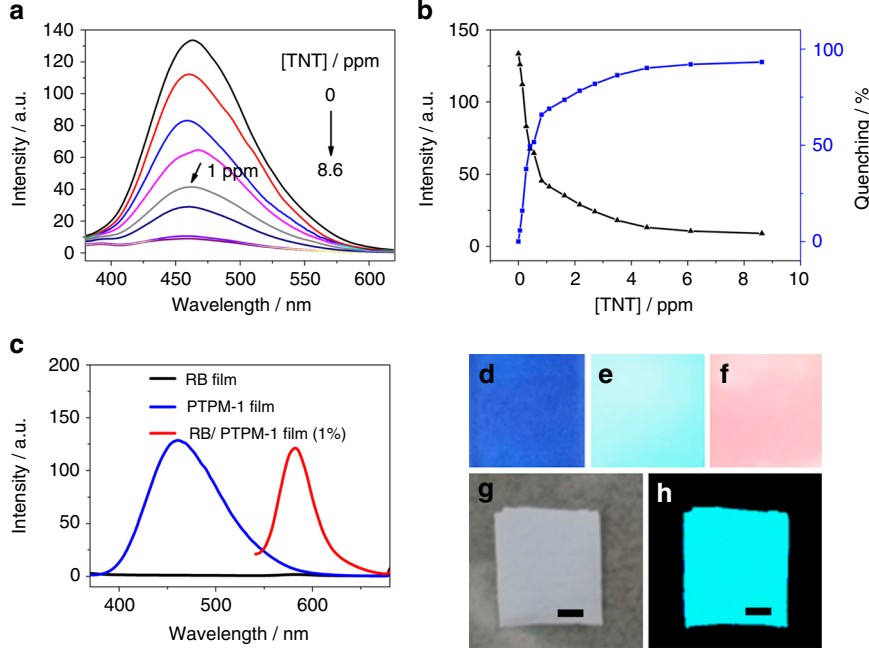

**Fig. 5** Applications of PTPM-1. **a** Emission spectra and **b** intensities and quenching ratio of PTPM-1 solution (0.02 mg/mL in 90% of water/THF mixtures (vol%)) upon adding of TNT (excited @346 nm); **c** emission spectra of RB film, PTPM-1 film, and RB/PTPM-1 film containing 1% (wt%) of RB (excited @346 nm); digital photos of **d** RB film, **e** PTPM-1 film and **f** RB/PTPM-1 film containing 1% (wt%) of RB (under irradiation with UV lamp @365 nm); digital photos of PTPM-1 textiles via static electrospinning: **g** under sunlight and **h** under irradiation with UV lamp @365 nm. Scale bar is 2.5 mm for all

PTPM to the well-known AIE molecule, tetraphenylethylene, the AIE mechanism of PTPM-1 should be due to the restriction of intermolecular rotations (RIR) theory, which has been widely investigated in AIE literatures.[23–25] At solution state, the rotation of these phenyl groups in PTPM-1 polymer chain is not restricted. At solid state, the RIR process causes the AIE behavior. In comparison, the rotation of meta-phenyl group of PTPM-2 is restricted at either solution state or solid state, resulting in ACQ

behavior. These phenomena also indicate that the rotation of phenyl groups of polymer main chain plays an important role in its AIE properties. In comparison with the widely studied AIE molecules with π-conjugated structures, the PTPMs are less conjugated, which is similar to literature results on heterodox luminogens containing no canonical chromophore, such as tetraphenylethane, oligosaccharide, and succinic anhydride moieties[23–25]. It is worth mentioning that PTPM-1 is a special type of

AIE polymer without any AIE-specific moieties. To reveal the photophysical properties of PTPM-1, a series of model compounds with high chemical similarity, i.e., triphenylmethanol, styryldiphenylmethanol, and polystyryldiphenylmethanol, was synthesized. The successful synthesis of small molecular model compounds, i.e., triphenylmethanol and styryldiphenylmethanol, is confirmed by [1]H NMR, [13]C NMR, and FT-IR spectra, and polystyryldiphenylmethanol is confirmed by [1]H NMR, [13]C NMR, FT-IR, and GPC results, as shown in Supplementary Figs. 18 and 19. Through the investigations on the photophysical properties of these model compounds (Supplementary Table 1 entry 1–3, and Supplementary Figs. 20–22), triphenylmethanol is weakly luminescent, but both styryldiphenylmethanol and polystyryldiphenylmethanol exhibit AIE behavior, which indicates that the triphenylmethanol group has characteristics of a chromophore, but only exhibits AIE behavior when conjugated with a vinyl group or incorporated in a polymer chain. These results confirm the AIE properties of PTPM-1, where triphenylmethanol is incorporated in the polymer chain. Since para-PTPM exhibits AIE property, it is the main focus of the following discussion. First, the AIE behavior of PTPM is examined by adding water to a THF solution of PTPM-1. When the water content is lower than 50%, the polymer solution is transparent with poor emission (Fig. 3a–c). As the water content increases above 50%, aggregates of polymer are formed with a gradual decrease of transmittance (Fig. 3c) and a gradual induction of photoluminescence intensity (Fig. 3a, b). The fluorescence quantum yield of the polymer solution also increases with water content (Fig. 3d). The AIE behavior of PTPM-1 can easily be further verified by addition of a drop of PTPM-1 solution onto a thin-layer chromatography plate (Fig. 3e). During the evaporation process of PTPM-1 solution, PTPM-1 starts to aggregate from the solution. Attributed to the AIE property of PTPM-1, the PTPM-1 spot on the thin-layer chromatography plate exhibits increasing photoluminescent intensity, as the solvent evaporates.

## Stimuli-responsive properties of PTPM-1.

Another intriguing aspect of PTPM-1 concerns the dual responsive properties of upper critical solution temperature (UCST)-type thermo-responsive behavior in polar organic solvents (DMSO, DMF, and THF), which is tunable over a wide temperature range with different amounts of water, and is accompanied by both turbidity and photoluminescence changes (Fig. 4). In DMSO, PTPM-1 exhibits tunable UCST from 37 to 88 °C as the water content increases from 23 to 31% (v/v). In DMF, its UCST value is tunable from 17 to 91 °C as the water content increases from 34 to 53% (v/v). In THF, its UCST can be tuned from 26 to 59 °C as the water content varies from 62 to 69% (v/v). This thermo-responsive behavior over a wide temperature range is similar to previous reports[26–28], where thermodynamic hydrogen bonds formation/deformation plays an important role, as illustrated in Supplementary Fig. 23. The trend in the water content needed for thermo-responsive behavior probably related to the capability of these solvents to form hydrogen bonds with hydroxide groups of PTPM-1 and water[26–28]. Taking PTPM-1 solution (2 mg/mL) in DMF with 45% of water (vol%) as an example, the UCST value is ~70 °C. Below the UCST, the polymer aggregates with high turbidity and emission intensity (Fig. 4b, c). As the temperature increases, the turbidity and emission intensity of the polymer solution decrease. The solution becomes transparent with poor emission when the temperature reaches 80 °C, i.e., above its UCST. Generally, dual thermo-responsive polymers are designed by incorporating chromophores into polymer chains[29, 30]. The PTPM reported here therefore offers another strategy of designing stimuli-responsive polymers.

## Potential applications of PTPM-1.

Final intriguing aspect of PTPM-1 discussed here relies on its potential applications in the areas of explosive detection, fluorescent dye doping, and luminescent textile preparation, as shown in Fig. 5. Attributed to the expected emission quenching caused by the charge–transfer interaction between electron deficient 2,4,6-trinitrotoluene (TNT) and electron rich PTPM-1, its potential application in explosive detection was tested. Upon addition of TNT at the ppm level, the emission intensity of PTPM-1 solution decreases rapidly (more than 70% of fluorescence quench ratio with 1 ppm of TNT), as shown in Fig. 5a, b. It therefore exhibits an excellent potential for application in explosive detection at the ppm level. Commercial available fluorescent dyes, e.g., rhodamine B (RB), are highly luminescent in dilute solution, while they are poorly luminescent because of self-quenching effect in solid state. Considering the urgent demands of solid-state luminescent materials in light-emitting diodes and luminescent sensors, to find a simple strategy for luminescence enhancement of commercial fluorescent dyes in their solid state is therefore considerably cost-effective. PTPM-1 is strong luminescent in its solid state. Its application in the luminescent enhancement of commercial fluorescent dyes in their solid state was therefore investigated by doping PTPM-1 film (host) with RB (guest) as an example. Even though solid-state RB film is poorly luminescent due to ACQ (Fig. 5c, d, the blue color in Fig. 5d is due to the reflection from the UV lamp at 365 nm), while the RB/PTPM-1 film containing only a tiny amount (1 wt %) of RB emits the luminescence of guest RB at 583 nm when excited at the excitation wavelength of AIE host PTPM-1 (346 nm) (Fig. 5c, f), which proves a simple strategy of luminescence enhancement of commercial ACQ fluorescent dye in solid state, by simply doping AIE polymer with tiny amount of readily available ACQ fluorescent dyes. This luminescent enhancement of RB is due to the inhibition of ACQ behavior of RB and the efficient fluorescence resonance energy transfer from AIE PTPM-1 to RB, when RB is well dispersed in PTPM-1 matrix. It is worth mentioning that this is also a simple strategy for emission of fluorescent dyes with enlarged Stokes shift upon excitation under UV light. Compared to small molecules, polymers exhibit distinguished processing properties and can be processed through spinning, injection molding, extrusion molding, etc., resulting in widespread polymeric materials with different shapes, purposes, and functionalities in our modern life. The advantage of the processing property of this AIE polymer over other small AIE molecules is finally tested by using electrospinning as an example. A luminescent textile made of PTPM-1 can be easily prepared by electrospinning of its concentrated solution in DMF, as illustrated in Fig. 5g, h.

In conclusion, the Barbier reaction has been successfully introduced to polymer synthesis. Through Barbier polyaddition, a series of monomers containing an organic halide and a benzoyl group (either AB type or A2+B2 type) have been successfully polymerized, resulting in the synthesis of a series of phenyl-methanol group containing polymers. It is worth mentioning that PTPM-1 is a special type of stimuli-responsive polymer in the area of dual thermo-responsive polymers (based on luminescence and turbidity) over a wide temperature range, and a special type of AIE polymer without any AIE moieties. The many advantages of using the Barbier reaction in polymer synthesis are as follows: its potential tolerance to functional groups, moisture and solvents;[19–21] the polymer is potentially degradable due to the reversibility of Barbier reaction; there is a wide range of monomers suitable for Barbier polyaddition; the outstanding AIE and thermo-responsive luminescent properties of Barbier polymer varying over a wide temperature range; and the potential sensory, luminescence enhancement of fluorescent dyes in solid state and processing properties of Barbier polymer. The

introduction of Barbier polyaddition expands the libraries of monomer and polymer in a way that opens an avenue in the design and application of functional polymer materials with both academic and economic perspectives.

## Methods

**Barbier polyaddition.** Typical procedure is as follows: to one flame-dried 2-neck round bottom flask containing 0.288 g (12.0 mmol) of freshly peeled Mg scraps was added 20 mL of dry THF. Then 2.167 g (10.0 mmol) 4-chlorobenzophenone dissolved in 10 mL THF was added to the flask at room temperature through a syringe. After stirring for 5 min, 0.1 mL of 1, 2-dibromoethane was added to the flask as activator. After the reaction was refluxed for 24 h, the solution was cooled to room temperature, followed by quenching and hydrolysis with 20 mL saturated aqueous ammonium chloride. After filtration and workup with dichloromethane/water, the organic solution was dried with anhydrous $MgSO_4$ and concentrated under reduced pressure. After the product was purified by precipitation into excessive petroleum ether, filtered, and dried under vacuum, 1.527 g PTPM-1′ was obtained as a yellow powder with a yield of 83.6%.

**Data availability.** The data that support the findings of this study are available within the article and its Supplementary Information file or from the corresponding author upon reasonable request.

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

## Acknowledgements

We acknowledge funding support from NSFC 51503226, Shandong Provincial NSF ZR2015EQ018 and ZR2016BQ32, China University of Petroleum (East China) starting funding and the Fundamental Research Funds for the Central Universities (Grand No. 15CX05028A).

## Author contributions

X.-L.S., D.-M.L., D.T., X.-Y.Z., W.W., and W.-M.W. performed the experiments and analyzed the data. W.-M.W. designed the study and wrote the manuscript.

## Additional information

**Competing interests:** The authors declare no competing financial interests.

