## [Peer Review File · Nature Communications]

Reviewers' comments:

Reviewer #1

Made comments to the editor only.

Reviewer #2 (Remarks to the Author):

In this manuscript, the authors introduced the Barbier reaction into polymer chemistry and aimed to develop a one-pot Barbier polyaddition method for the construction of a series of phenylmethanol group containing polymers. The AIE and thermo-responsive properties of para-triphenylmethanol-containing polymer were investigated. The solid-state luminescence of PTPM-1 is unique as its structure shows limited conjugation. Also, it is a new type of AIE polymer without any AIE moieties. This work is of great interest. The paper may be accepted for publication in Nature Communications after careful revision.

1. The authors should check the whole manuscript very carefully. There exist several serious mistakes in the manuscript. For example, according to Table S1 and the discussion in the text, the unit of Mn in Table 1 should be Da, not kDa. In Figure 3C, the legend should be PTPM-1 film and RB/PTPM-1 film, not PDPM. PDPM and PTPM-1 are two totally different polymers in this work.

2. The authors only show the ¹H NMR spectra of the obtained polymers to support their claim that they indeed get polymers with expected structures. No detailed analysis of the ¹H NMR spectra was given in the text. The polymer structures given in Table S1 should be fully characterized. The authors should supplement the FT-IR spectra and carbon NMR spectra of all the obtained polymers and provide the corresponding analysis and peak assignment of the IR and NMR results in the manuscript and supporting information.

3. Why choose TPM, SDPM and PSDPM as model compounds to study the photophysical properties of PTPM, PDPM or PMPM? What is the relationship between these two systems? The authors should provide discussions about how to use the photophysical properties of the model compounds to explain the photophysical behaviors of PTPM, etc.

4. The emphasis point of this paper is not clear to the reader. The authors stated that "Detailed investigations on the polymerization mechanism will be discussed elsewhere but the discussion below focuses on some key properties of the polymers synthesized using the Barbier reaction." However, the title, abstract, introduction and conclusion indicated that the development of new polymerization method is an important part in this paper. Why not discuss the polymerization mechanism in this manuscript?

5. Although the photophysical properties of the polymers are discussed in detail in this manuscript, no deep investigation or explanation of the underlying mechanism was given. For example, what is the AIE mechanism of PTPM-1 and ACQ mechanism of PTPM-2 from the structural point of view? Why PTPM-1 and PTPM-2 show an opposite photophysical behavior? The structure of PDPM is similar to that of PTPM. Why does PDPM show weak fluorescence?

6. The authors stated that the conversion in Table 1 and Table S1 was calculated from ¹H NMR spectroscopy. However, no detailed calculation method was provided.

7. The statement of "a series of monomers have been selected with the benzene rings as side groups (Supplementary Table S1 #5-8), which should favor polymerization rather than cyclization in the presence of rigid benzene rings" is confusing. What is the meaning of benzene rings as side groups in the monomer structures in Table S1 #5-8? In addition, why the monomer structure in Table S1 #6 can overcome the potential intramolecular cyclization?

8. Figure S1 was not mentioned or discussed in the manuscript.

9. The narrow PDI and unimodal and symmetric GPC curves cannot determine that this polymerization is a chain-growth polymerization rather than step-growth polymerization. Further evidence should be provided to support the chain-growth hypothesis. Besides, the GPC curves in Figure S10 seems not complete. What happens after elution time of 10 mL?

10. In the procedure of Barbier polyaddition, why 0.1 mL of 1,2-dibromoethane was added into the reaction system? Besides, some of the ¹H NMR spectra of the obtained polymer were shown from 2 or 3 ppm. Please provide a full ¹H NMR spectra of the polymers including the region below 2 or 3

ppm.

11. The authors stated that using Barbier reaction in polymer synthesis has the advantage of tolerance to moisture. But in the polymerization process, distilled THF was used. Water was added to quench the reaction. What is the supporting evidence for the claim that this reaction is tolerant to moisture?

12. Why can the emission of RB be enhanced by doping PTPM-1? What is the underlying mechanism? Please discuss it in the manuscript.

13. The photo of TPM in Figure S12 was taken under 365-nm UV light and show a deep-blue luminescence. But from its emission spectrum, it should not exhibit visible emission. Please supplement the emission spectrum of TPM excited @ 365 nm. If the emission maximum of TPM is different when excited under different wavelength, in other words, if TPM can show excitation-dependent emission property, please provide corresponding discussion and explanation in the manuscript.

14. If possible, please supplement the absorption spectra of the solid thin films of TPM, SDPM, PTPM-1 and PSDPM, and supplement the excitation spectrum of the PSDPM solid. Meanwhile, please compare their respective absorption spectra and excitation spectra in the solid state. It seems that the absorption spectra and the excitation spectra of the solution of TPM, SDPM, PTPM-1, and PSDPM are very different, why?

15. As shown in Table S2, the quantum yield of the solution of SDPM, PSDPM and PTPM-1 are higher than the corresponding quantum yield of their solid. How to explain such contradiction?

16. The authors should add more data points in Figure 1 to show a detailed variation tendency of PTPM-1 with the change of water fraction. For example, what is the situation when fw is 10%, 20%, 30% or 40%?

17. Why is 346 nm chosen as the excitation wavelength of PTPM-1 in Figure 1–3? What is the meaning to measure the emission spectra of the solution of the polymer under different excitation wavelength in Figure S12–S16 A and B?

18. To make this manuscript more readable, it is suggested to place the supporting figures and tables in the order that they appear in the text. Table S1 is frequently mentioned and discussed in the manuscript and the content in Table 1 is totally included in Table S1, so it is suggested to replace Table 1 with Table S1 in the manuscript.

19. Some figures in this manuscript miss necessary information. For example, there is no units in the Y-axis of Figure 1A–C, 2A–B, 3A–C and Figure S12–21. What size does the scale bar in Figure 3G and H stands for? The meaning of fw and ΦF in Figure 1 and the meaning of the parameters in Table S2 should be provided in the corresponding Figure/Table caption. The solvent type used in the measurement of results in Table S2 and Figure S12–21 should be given.

Reviewer #3 (Remarks to the Author):

The original novelty of this study is use of Barbier reaction to synthesize the polymers with interesting AIE and stimuli-responsive properties although the Barbier reaction has been known in organic chemistry for more than a century. Thus, the Figures about preparation and characterizations of the polymers via Barbier reaction should be put in the main text, not in Supplementary information as shown in this manuscript. More evidences are required such as, identification of terminal groups in order to support the proposed polymerization mechanism shown in Scheme 1. Even though this manuscript is not well-organized and shortcoming of this manuscript is obvious, I strongly recommend accepting this manuscript after careful revision because this study opens a new route to synthesize a series of new polymers, which should be interested by extensive readers. Additional comments are as follows.

1. All GPC traces in Fig. S2~S10 show a shoulder (or peak) at low molecular weight direction. In addition, the GPC curves should be in the range of 5~13 mL/min in order to display components of the formed polymers because the polymers obtained from step polymerization generally display multiple GPC curves except those polymers obtained via full precipitation separation, narrow and single GPC curves are unusual.

2. "After stirring for 5 min, 0.1 mL of 1, 2-dibromoethane was added to the flask", the purpose for addition of 1, 2-dibromoethane into the reaction solution should be clearly explained.

3. Page 4, line 65-67, "It is worth mentioning that molecular weight distributions of the synthesized polymers are relatively narrow, with PDI smaller than 1.5, with a feature of chain-growth polymerization, rather than step-growth polymerization, which is shown by the unimodal and symmetric GPC curves". As discussed in the text, this is a polyaddition reaction of AB or A2 + B2 monomer, which determines that this is a step polymerization, not chain polymerization because in the polymerization, the reactions between the AB monomers, or oligomers with A and B groups respectively at their ends should occur. Obviously, the GPC curves shown in Fig.S2~Fig.S10 are artificially selected.

4. In the ¹H NMR spectra of Fig.2~Fig.9, the hydroxyl proton signal should be marked; and the signal(s) of phenyl proton(s) next to the carbonyl group seem to appear in the ¹H NMR spectra of the polymers, which refers to the monomers remained in the polymers, or to the terminal units. If the latter case is correct, the M_n can be calculated from the ¹H NMR data.

5. Note in Table 1 and S1 says that conversions were calculated based on the proton NMR data, based on the ¹H NMR spectra in Fig.S2~S9, this calculation may have big error because ¹H NMR spectra of the monomers are overlapped with that of their corresponding polymers. It is best to give the yields.

6. It is better to merge Table 1 and Table S1 because Table S1 includes many data listed in Table 1.

7. "Figure S15..... A) absorption spectrum, emission spectrum (excited @ 259 nm) of solution, B) excitation spectrum, emission spectrum (excited @ 274nm) of solution and digital photos (under sunlight and irradiation with UV lamp @ 365 nm)". "digital photos (under sunlight and irradiation with UV lamp @ 365 nm)" is in Fig.S15A, not in Fig.15B, please check and revise!

8. In Aggregation-induced emission (AIE) measurements section of the supporting information, "The fluorescence spectra and transmittance at 500 nm of polymers in THF/Water mixtures with different water content were recorded respectively on a Hitachi F2500 fluorescence spectrofluorometer and on a Shimadzu UV-2450 UV-Vis spectrophotometer", Water content in the THF/water mixture did not affect the size of polymer aggregates (all are 500 nm), this is not reasonable.

9. Page 5, line 89-90, "During the evaporation process of PTPM-1 solution, its photoluminescence is gradually induced". How was the conclusion "its photoluminescence is gradually induced" drawn based on Fig.1E? The phenomenon in Fig.1E should be clearly explained in order to get "its photoluminescence is gradually induced".

10. There is no number in scale bar in Fig.3G and 3H.

We thank the reviewers for the constructive comments, address each comment below and highlight where the changes were made in manuscript with yellow color. Because there are a lot of revised figures, we therefore do not provide all of the revised figures in this cover letter. Corresponding changes have been made and highlighted in the manuscript and supporting information.

Reviewer #1

Made comments to the editor only.

Comment from the editor: Reviewer #1 only made comments to the editor, but suggests an interest in publication.

Response: Thanks for the suggestion of accepting our manuscript for publication and the efforts on our manuscript.

Reviewer #2

Comments:

In this manuscript, the authors introduced the Barbier reaction into polymer chemistry and aimed to develop a one-pot Barbier polyaddition method for the construction of a series of phenylmethanol group containing polymers. The AIE and thermo-responsive properties of para-triphenylmethanol-containing polymer were investigated. The solid-state luminescence of PTPM-1 is unique as its structure shows limited conjugation. Also, it is a new type of AIE polymer without any AIE moieties. This work is of great interest. The paper may be accepted for publication in Nature Communications after careful revision.

Response: Thanks for the recommendation of accepting our manuscript for publication and the efforts on our manuscript. We have revised our manuscript according to reviewer's comments. The point-by-point responses have been listed as follows:

1. The authors should check the whole manuscript very carefully. There exist several serious mistakes in the manuscript. For example, according to Table S1 and the discussion in the text, the unit of Mn in Table 1 should be Da, not kDa. In Figure 3C, the legend should be PTPM-1 film and RB/PTPM-1 film, not PDPM. PDPM and PTPM-1 are two totally different polymers in this work.

Response: Thanks for pointing out these mistakes in our manuscript. We have changed the unit of Mn in Table 1 to Da and the legend of Figure 3C. We have double checked the whole manuscript carefully and revised the mistakes in our manuscript according to reviewer's comments, highlighted with yellow color.

2. The authors only show the ^1H NMR spectra of the obtained polymers to support their claim that they indeed get polymers with expected structures. No detailed analysis of the ^1H NMR spectra was given in the text. The polymer structures given in Table S1 should be fully characterized. The authors should supplement the FT-IR spectra and carbon NMR spectra of all the obtained polymers and provide the corresponding analysis and peak assignment of the IR and NMR results in the manuscript and supporting information.

Response: Thanks for these great suggestions. Detailed analysis and peak assignment of the ^1H NMR spectra have been supplemented in the experimental section and revised figures. Corresponding analysis has been added in the revised manuscript. Meanwhile, we have supplemented FT-IR, carbon NMR, XPS and solid UV-vis tests to fully characterize all of the obtained polymers, according to reviewer's suggestions, which are very helpful to understand the end group of polymers and the corresponding polymerization mechanism. Almost all of the figures in supporting information have been revised by adding FT-IR, carbon NMR, XPS and solid UV-vis spectra with corresponding peak assignment. The analysis of these characterizations and corresponding discussion of polymerization mechanism have been added in the manuscript, as follows: "Taking the Barbier polymerization of 4-bromobenzophenone monomer in the synthesis of PTPM-1 polymer as an example, the

successful polymer synthesis is verified by the disappearance of sharp ^1H NMR signals of monomer (Fig. 1A) and the appearance of broader aromatic signals at $\sim 7.5\text{--}7.0$ ppm and hydroxide proton at ~ 3.0 ppm (Fig. 1B). The successful formation of polymer is further confirmed by GPC characterization with a M_n of 4700 and PDI of 1.28 (Fig. 1C). From the terminal benzophenone group existence (~ 7.8 ppm in Fig. 1B), carbonyl group existence (verified by FT-IR and XPS spectra in Fig. 1D and Fig. 1E, respectively) and the appearance of C-OH group (verified by ^1H NMR, FT-IR and XPS spectra in Fig. 1B, Fig. 1D and Fig. 1E, respectively), the polymerization mechanism is verified as Barbier polyaddition. For a better understanding of Barbier reaction in polymerization, the polymerization mechanism is illustrated in Scheme 1 (This work), via both AB type and A2+B2 type Barbier polyaddition between Grignard reagent and carbonyl group in one pot. As the Grignard reagent forms, the addition of highly reactive Grignard reagent to the carbonyl group takes place successively in one pot. Because of the bifunctionality of the monomers, either AB type or A2+B2 type, continuous addition reactions between Grignard reagents and carbonyl groups happen, i.e., polyaddition mechanism, resulting in successful polymer synthesis. Other examples of chemical structures of monomers and the corresponding polymers synthesized via Barbier polyaddition are listed in Table 1. From the comparison of ^1H NMR spectra of these monomers and corresponding products (Supplementary Fig. S1-S8), the products clearly exhibit much broader signals for aromatic protons at $\sim 8.0\text{--}6.5$ ppm, hydroxide proton at $\sim 3.5\text{--}2.5$ ppm and alkyl protons at $\sim 2.5\text{--}1.0$ ppm, which is typical evidence of successful polymer synthesis.”.

3. Why choose TPM, SDPM and PSDPM as model compounds to study the photophysical properties of PTPM, PDPM or PMPM? What is the relationship between these two systems? The authors should provide discussions about how to use the photophysical properties of the model compounds to explain the photophysical behaviors of PTPM, etc.

Response: TPM, SDPM and PSDPM are triphenylmethanol, triphenylmethanol containing monomer, triphenylmethanol containing polymer respectively, which are the simplest triphenylmethanol compounds. The reason that we choose these compounds as model compounds to study the photophysical properties of our polymers, is due to the chemical similarity between these two systems. The study of the photophysical properties of model compounds can confirm that the AIE behavior of PTPM is derived from the photophysical properties of triphenylmethanol moieties. According to reviewer’s comments, we have added the reason why to choose these model compounds, and have revised the manuscript with corresponding discussions, as follows: “To reveal the photophysical properties of PTPM-1, a series of model compounds with high chemical similarity, i.e., triphenylmethanol, styryldiphenylmethanol and polystyryldiphenylmethanol, were synthesized. The successful synthesis of small molecular model compounds, i.e., triphenylmethanol and styryldiphenylmethanol are confirmed by ^1H NMR, ^{13}C NMR and FT-IR spectra, and polystyryldiphenylmethanol are confirmed by ^1H NMR, ^{13}C NMR, FT-IR and GPC results, as shown in Fig. S18 and S19. Through the investigations on the photophysical properties of these model compounds (Supplementary Table S1 #1-3, and Supplementary Fig. S20-22), triphenylmethanol is weakly luminescent, but both styryldiphenylmethanol and polystyryldiphenylmethanol exhibit AIE behavior, which indicates that the triphenylmethanol group has characteristics of a chromophore, but only exhibits AIE behavior when conjugated with a vinyl group or incorporated in a polymer chain. These results confirm the AIE properties of PTPM-1, where triphenylmethanol is incorporated in the polymer chain.”.

4. The emphasis point of this paper is not clear to the reader. The authors stated that “Detailed investigations on the polymerization mechanism will be discussed elsewhere but the discussion below focuses on some key properties of the polymers synthesized using the Barbier reaction.” However, the title, abstract, introduction and conclusion indicated that the development of new polymerization method is an important part in this paper. Why not discuss the polymerization mechanism in this manuscript?

Response: Sorry for making reviewer confused. The polymerization mechanism of this Barbier type polymerization is polyaddition mechanism. And the “Detailed investigation on the polymerization mechanism” is just one hypothesis. We raise this hypothesis that this polyaddition exhibits a feature of chain-growth

polyaddition, because of narrow molecular weight distribution, which doesn't affect the polyaddition mechanism mentioned above. Similarly, reviewer #3 mentioned this hypothesis as well. After careful thinking, we realize that this hypothesis is wrong. Previously, we thought the discussion of this hypothesis will be discussed elsewhere. Right now, this hypothesis doesn't need any discussion any more. To eliminate this misunderstanding, we have enhanced the polyaddition mechanism and revised the manuscript as follows: "From the terminal benzophenone group existence (~7.8 ppm in Fig. 1B), carbonyl group existence (verified by FT-IR and XPS spectra in Fig. 1D and Fig. 1E, respectively) and the appearance of C-OH group (verified by ¹H NMR, FT-IR and XPS spectra in Fig. 1B, Fig. 1D and Fig. 1E, respectively), the polymerization mechanism is verified as Barbier polyaddition. For a better understanding of Barbier reaction in polymerization, the polymerization mechanism is illustrated in Scheme 1 (This work), via both AB type and A2+B2 type Barbier polyaddition between Grignard reagent and carbonyl group in one pot. As the Grignard reagent forms, the addition of highly reactive Grignard reagent to the carbonyl group takes place successively in one pot. Because of the bifunctionality of the monomers, either AB type or A2+B2 type, continuous addition reactions between Grignard reagents and carbonyl groups happen, i.e., polyaddition mechanism, resulting in successful polymer synthesis."

5. Although the photophysical properties of the polymers are discussed in detail in this manuscript, no deep investigation or explanation of the underlying mechanism was given. For example, what is the AIE mechanism of PTPM-1 and ACQ mechanism of PTPM-2 from the structural point of view? Why PTPM-1 and PTPM-2 show an opposite photophysical behavior? The structure of PDPM is similar to that of PTPM. Why does PDPM show weak fluorescence?

Response: Thanks for these great comments, which are very helpful for us to understand the photophysical properties of these polymers. As the chemical structures of these phenylmethanol containing polymers are concerned, when the amounts of phenyl groups increases, the photophysical properties of these polymers increase, indicating the importance of phenyl groups on their photophysical properties. Similar to the well-known AIE moiety with multiple phenyl groups, tetraphenylethylene, PTPM-1 contains three phenyl groups on one carbon center. The AIE mechanism of PTPM-1 should be due to the well-known restriction of intermolecular rotations (RIR). The different photophysical behavior of PTPM-1 and PTPM-2 should be due to the fact that the rotation of para-phenyl ring of PTPM-1 polymer main chain is not restricted in solution and restricted at solid state, while the rotation of meta-phenyl ring of PTPM-2 polymer main chain is restricted at both solution state and solid state. As the decrease of the amounts of phenyl groups on carbon center, the steric hindrance will decrease, resulting in the decrease of the restriction of rotation of phenyl rings and photophysical properties. PDPM therefore shows weak fluorescence.

According to reviewer's comments, we have therefore revised the manuscript by adding the deep explanation of the underlying mechanism and the comparison of photophysical properties of PTPM-1 and PTPM-2, as follows: "Obviously, the photophysical properties of these phenylmethanol polymers benefit from the increase of the amounts of phenyl groups on carbon center of methanol. Considering the chemical similarity of triphenylmethanol moieties of PTPM to the well-known AIE molecule, tetraphenylethylene, the AIE mechanism of PTPM-1 should be due to the restriction of intermolecular rotations (RIR) theory, which has been widely investigated in AIE literatures.²³⁻²⁵ At solution state, the rotation of these phenyl groups in PTPM-1 polymer chain is not restricted. At solid state, the RIR process causes the AIE behavior. In comparison, the rotation of meta-phenyl group of PTPM-2 is restricted at either solution state or solid state, resulting in ACQ behavior. These phenomena also indicate that the rotation of phenyl groups of polymer main chain plays an important role in its AIE properties."

6. The authors stated that the conversion in Table 1 and Table S1 was calculated from ¹H NMR spectroscopy. However, no detailed calculation method was provided.

Response: Just as commented by Review #3, "the calculation of conversion might have big error because ¹H NMR spectra of the monomers are overlapped with that of their corresponding polymers. It is best to give the

yields.” We therefore delete the conversions and just give yields, according to reviewers’ comments.

7. The statement of “a series of monomers have been selected with the benzene rings as side groups (Supplementary Table S1 #5-8), which should favor polymerization rather than cyclization in the presence of rigid benzene rings” is confusing. What is the meaning of benzene rings as side groups in the monomer structures in Table S1 #5-8? In addition, why the monomer structure in Table S1 #6 can overcome the potential intramolecular cyclization?

Response: Sorry for this confusing statement. What we wanted to describe in the previous manuscript is that the rigid benzene ring with steric hindrance effect is better to overcome the potential intramolecular cyclization than that with smaller steric hindrance groups. So we choose monomers with phenyl groups. Polymer would be synthesized with phenyl group either in the main chain or as side group of the resulting polymer (not in the monomer). We therefore revised the manuscript as follows: “To overcome the potential intramolecular cyclization, a series of monomers have been selected with halide and carbonyl groups in the para- (Table 1 #1, #2 and #4) or meta-positions (Table 1 #3) of one rigid benzene ring, or with the benzene rings as steric hindrance groups (Table 1 #5-8). The steric hindrance effect of phenyl groups either in the main chain or as side groups of the resulting polymers would favor polymerization rather than cyclization in the presence of rigid benzene rings.”. Even though the monomer in Table 1 #6 has ortho-phenyl group, both of the substituents are bromide. After forming Grignard reagents on ortho position of phenyl group, they will react with carbonyl groups on the other molecule, rather than itself. So, the potential intramolecular cyclization is overcome.

8. Figure S1 was not mentioned or discussed in the manuscript.

Response: According to reviewer’s comment, we have added the discussion of Figure S1 (Figure S18 in the revised supporting information) in the revised manuscript, as follows: “The successful synthesis of small molecular model compounds, i.e., triphenylmethanol and styryldiphenylmethanol are confirmed by ^1H NMR, ^{13}C NMR and FT-IR spectra, and polystyryldiphenylmethanol are confirmed by ^1H NMR, ^{13}C NMR, FT-IR and GPC results, as shown in Fig. S18 and S19.”.

9. The narrow PDI and unimodal and symmetric GPC curves cannot determine that this polymerization is a chain-growth polymerization rather than step-growth polymerization. Further evidence should be provided to support the chain-growth hypothesis. Besides, the GPC curves in Figure S10 seems not complete. What happens after elution time of 10 mL?

Response: Thanks for pointing out this improper hypothesis. This hypothesis doesn’t affect the Barbier polyaddition mechanism mentioned above. We therefore revised the manuscript by deleting this hypothesis, as follows: “It is worth mentioning that molecular weight distributions of the synthesized polymers are relatively narrow (PDI < 1.5) (Supplementary Fig. S9).”. Our GPC system is equipped with one TSK-Gel GMH_{HR}-N column, the signal after elution volume of 10 mL should be due to oligomers and then solvent. We therefore only reported results before 10 mL in the previous manuscript. According to reviewer’s comment, we have provided all of GPC curves in the range of 5~13.5 mL for better illustration of the results. Because there are a lot of GPC results revised, we therefore do not provide the revised GPC results in this cover letter. Corresponding changes have been made and highlighted in the supporting information.

10. In the procedure of Barbier polyaddition, why 0.1 mL of 1,2-dibromoethane was added into the reaction system? Besides, some of the ^1H NMR spectra of the obtained polymer were shown from 2 or 3 ppm. Please provide a full ^1H NMR spectra of the polymers including the region below 2 or 3 ppm.

Response: As is known, 1,2-dibromoethane can act as an activator in the Barbier reaction and Grignard reaction. Actually, 1,2-dibromoethane is not necessary in our research. However, the Grignard reagent formation is sluggish and affected by halide type and substituents. To reduce the initial reactivity differences between monomers containing different halides and substituents, we therefore chose 1,2-dibromoethane as the activating agent to initiate the sluggish Grignard reactions in the experiments. According to reviewer’s comment, we have

added the explanation in the experimental section, as follows: “0.1 mL of 1, 2-dibromoethane was added to the flask as activator.”. We have provided full ^1H NMR spectra of all polymers from 0 ppm in the revised manuscript according to reviewer’s comments. Because there are a lot of ^1H NMR spectra revised, we therefore do not provide the revised spectra in this cover letter. Corresponding changes have been made and highlighted in the supporting information.

11. The authors stated that using Barbier reaction in polymer synthesis has the advantage of tolerance to moisture. But in the polymerization process, distilled THF was used. Water was added to quench the reaction. What is the supporting evidence for the claim that this reaction is tolerant to moisture?

Response: We used the classic Barbier reaction in this work, where water has to be removed. However, the modified Barbier reaction has the potential to be used with water or even in water (Chao-Jun Li* and Wen-Chun Zhang, Unexpected Barbier-Grignard Allylation of Aldehydes with Magnesium in Water, *J. Am. Chem. Soc.* 120, 9102-9103 (1998); Feng Zhou and Chao-Jun Li*, The Barbier-Grignard-type arylation of aldehydes using unactivated aryl iodides in water. *Nat. Commun.* 5, 4254 (2014)). We therefore stated that the Barbier polyaddition has the potential advantage of tolerance to moisture. This raised comment by reviewer is very interesting. Further utilization of this Barbier polyaddition in aqueous condition, with functional groups and by different metals will be discussed elsewhere. To eliminate the misunderstanding, “tolerance” has been changes to “potential tolerance” and references have been added to this statement, as follows: “Its potential tolerance to functional groups, moisture and solvents,¹⁹⁻²¹”.

12. Why can the emission of RB be enhanced by doping PTPM-1? What is the underlying mechanism? Please discuss it in the manuscript.

Response: The mechanism of emission enhancement of RB by PTPM-1 is fluorescence resonance energy transfer. The host PTPM-1 film exhibits an maximum emission at 465 nm, where the guest RB has absorption. When the RB/PTPM-1 film containing only a tiny amount (1 wt%) of RB was prepared through evaporation, the aggregation caused quenching of RB is inhibited. The efficient fluorescence resonance energy transfer from PTPM-1 to RB will happen in the solid state, resulting in emission enhancement of RB. We have revised the corresponding discussion as follows: “Even though solid-state RB film is poorly luminescent due to ACQ (Fig. 3C and 3D, the blue color in Fig. 3D is due to the reflection from the UV lamp at 365 nm), while the RB/PTPM-1 film containing only a tiny amount (1 wt%) of RB emits the luminescence of guest RB at 583 nm when excited at the excitation wavelength of AIE host PTPM-1 (346 nm) (Fig. 3C and 3F), which proves a simple strategy of luminescence enhancement of commercial ACQ fluorescent dye in solid state, by simply doping AIE polymer with tiny amount of readily available ACQ fluorescent dyes. This luminescent enhancement of RB is due to the inhibition of ACQ behavior of RB and the efficient fluorescence resonance energy transfer from AIE PTPM-1 to RB, when RB is well-dispersed in PTPM-1 matrix.”.

13. The photo of TPM in Figure S12 was taken under 365-nm UV light and show a deep-blue luminescence. But from its emission spectrum, it should not exhibit visible emission. Please supplement the emission spectrum of TPM excited @ 365 nm. If the emission maximum of TPM is different when excited under different wavelength, in other words, if TPM can show excitation-dependent emission property, please provide corresponding discussion and explanation in the manuscript.

Response: The blue luminescence phenomenon under 365 nm UV light is just due to the reflection from the UV lamp at 365 nm, same as the result in Fig. 3D. TPM solids reflect more light than the background. This is just due to the fact the TPM solids are crystalized powders. Actually, this reflected light is very weak, but can still be detected by the camera. According to reviewer’s comments, we have monitored the emission spectra of TPM (excited @ 260, 268, 280, 300, 320, 340, 365, 380 and 400 nm respectively), as shown in Figure A, which shows that TPM has no excitation-dependent emission property.

Figure A. The emission spectra of TPM (excited @ 260, 268, 280, 300, 320, 340, 365, 380 and 400 nm respectively).

14. If possible, please supplement the absorption spectra of the solid thin films of TPM, SDPM, PTPM-1 and PSDPM, and supplement the excitation spectrum of the PSDPM solid. Meanwhile, please compare their respective absorption spectra and excitation spectra in the solid state. It seems that the absorption spectra and the excitation spectra of the solution of TPM, SDPM, PTPM-1, and PSDPM are very different, why?

Response: We have supplemented the solid absorption spectra of all the samples (not limited to TPM, SDPM, PTPM-1 and PSDPM) and the excitation spectrum of the PSDPM solid in supporting information, according to reviewer's comments. TPM, SDPM, PTPM-1 and PSDPM have different absorption spectra and excitation spectra in the solid state, which might be due to the particularity of AIE materials. As absorption spectra and excitation spectra of these molecules in the solution state are concerned, their main peaks look similar (SDPM as an example in Figure B). Thanks for pointing out this strange phenomenon. We have tried literature research and found no explanation of this phenomenon. This is worthy of investigation. We have started the investigation on the photophysical properties of our polymers by quantum chemistry calculation. We hope we can find a reliable explanation and will try to explain this phenomenon well elsewhere.

Figure B. Absorption and excitation spectra of SDPM in the solution.

15. As shown in Table S2, the quantum yield of the solution of SDPM, PSDPM and PTPM-1 are higher than the corresponding quantum yield of their solid. How to explain such contradiction?

Response: The luminescence of AIE compounds relies on the restriction of intermolecular rotations (RIR) process. The RIR process should highly affect the luminescence efficiency of AIE compounds. In comparison to the luminescence in solution, the luminescence efficiency in solid/aggregation state will be affected by both chemical structure and aggregation status. A lot of luminescent compounds exhibit quantum yield higher than 80% in solution, while many AIE compounds exhibit low quantum yield of ~10% (for example: Figure C from Hong, Y., Lam, J. W. Y. & Tang, B. Z. Aggregation-induced emission. Chem. Soc. Rev. 40, 5361-5388 (2011)). So, this phenomenon might be due to the lower luminescence efficiency of AIE compounds in solid/aggregation state than that in solution. Thanks for pointing out this strange phenomenon. We have started the investigation on the photophysical properties of our polymers by quantum chemistry calculation. We hope we can find a reliable explanation and will try to explain this phenomenon well elsewhere.

Fig. 2 Fluorescence photographs of the solutions or suspensions of the molecules or nanoaggregates of **1** in THF/water mixtures with different volumetric fractions of water (f_w , vol%); fluorescence quantum yields (Φ_F , %) of **1** estimated using quinine sulfate as standard. Reproduced with permission from ref. 30. Copyright (2008) Wiley-VCH.

Figure C. Example of AIE compound exhibiting low quantum yield of ~10%.

16. The authors should add more data points in Figure 1 to show a detailed variation tendency of PTPM-1 with the change of water fraction. For example, what is the situation when f_w is 10%, 20%, 30% or 40%?

Response: We have add more data points (10%, 20%, 30% and 40%) in Figure 1 (Figure 2 in the revised manuscript) to show a detailed variation tendency of PTPM-1 with the change of water fraction according to reviewer's comments.

Figure 2. Aggregation-induced emission properties of PTPM-1. A) Emission spectra (excited @346 nm), B) emission intensities (excited @346 nm), C) transmittance @500 nm wavelength and digital photos (under sunlight and irradiation with UV lamp @365 nm), and D) fluorescent quantum yields (Φ_F) of PTPM-1 (0.1 mg/mL) in water/THF mixtures with different water volume fractions (f_w); E) digital photos of one drop of PTPM-1 solution (10 mg/mL in THF) on thin layer chromatography plate with different evaporation timescale at room temperature (under irradiation with UV lamp @ 365 nm).

17. Why is 346 nm chosen as the excitation wavelength of PTPM-1 in Figure 1-3? What is the meaning to measure the emission spectra of the solution of the polymer under different excitation wavelength in Figure S12-S16 A and B?

Response: Generally, the optimized wavelength from excitation spectrum is chosen as the excitation wavelength. For each compound at one status (either soluble solution, aggregation solution or solid film), once the excitation wavelength is chosen, the excitation wavelength is kept the same so that the comparison of emission intensity is meaningful. For the aggregation solution of PTPM-1, the optimal excitation wavelength is 346nm. So, 346 nm have been kept the same and used in Figure 1-3 (Figure 2-4 in the revised manuscript). The excitation wavelengths in Figure S12-S16 A (Figure S11-S12 and S20-S22 A in the revised manuscript) are based on maximum wavelength from absorption spectra, while those in Figure S12-S16 B (Figure S11-S12 and S20-S22 B in the revised manuscript) are based on the maximum wavelength from excitation spectra. These maximum wavelengths based on the absorption and excitation spectra are a little bit different. This is the reason that we chose different excitation wavelength in Figure S12-S16 A and B. However, there are no differences from

emission spectra.

18. To make this manuscript more readable, it is suggested to place the supporting figures and tables in the order that they appear in the text. Table S1 is frequently mentioned and discussed in the manuscript and the content in Table 1 is totally included in Table S1, so it is suggested to replace Table 1 with Table S1 in the manuscript.

Response: Thanks for this great suggestion. In revised manuscript, we have placed the supporting figures in the order that they appear in the text, and replaced Table 1 with Table S1 according to reviewer's comments.

19. Some figures in this manuscript miss necessary information. For example, there is no units in the Y-axis of Figure 1A-C, 2A-B, 3A-C and Figure S12-21. What size does the scale bar in Figure 3G and H stands for? The meaning of fw and ΦF in Figure 1 and the meaning of the parameters in Table S2 should be provided in the corresponding Figure/Table caption. The solvent type used in the measurement of results in Table S2 and Figure S12-21 should be given.

Response: Thanks for pointing these information out. We have supplemented these miss necessary information and highlighted them with yellow color, according to reviewer's comments.

Reviewer #3

The original novelty of this study is use of Barbier reaction to synthesize the polymers with interesting AIE and stimuli-responsive properties although the Barbier reaction has been known in organic chemistry for more than a century. Thus, the Figures about preparation and characterizations of the polymers via Barbier reaction should be put in the main text, not in Supplementary information as shown in this manuscript. More evidences are required such as, identification of terminal groups in order to support the proposed polymerization mechanism shown in Scheme 1. Even though this manuscript is not well-organized and shortcoming of this manuscript is obvious, I strongly recommend accepting this manuscript after careful revision because this study opens a new route to synthesize a series of new polymers, which should be interested by extensive readers. Additional comments are as follows.

Response: Thanks for the strong recommendation of accepting our manuscript for publication and the efforts on our manuscript. We have added the preparation and characterizations of one representative polymer (PTPM-1) as Figure 1 in the main text.

Figure 1. ^1H NMR spectra of 4-bromobenzophenone (A) and PTPM-1 (B) in CDCl_3 , GPC curve of PTPM-1 (C) in THF, FT-IR spectrum of PTPM-1 (D) and XPS $\text{O}1s$ spectra of PTPM-1 (E).

Thanks for the great suggestions on the polymerization mechanism, which are very helpful. Terminal group identification has been carried out by NMR, FT-IR and XPS tests, which supports the proposed polymerization mechanism shown in Scheme 1. We have revised the manuscript by adding the analysis of characterizations and the discussion of corresponding polymerization mechanism, as follows: “Taking the Barbier polymerization of 4-bromobenzophenone monomer in the synthesis of PTPM-1 polymer as an example, the successful polymer synthesis is verified by the disappearance of sharp ^1H NMR signals of monomer (Fig. 1A) and the appearance of broader aromatic signals at $\sim 7.5\text{--}7.0$ ppm and hydroxide proton at ~ 3.0 ppm (Fig. 1B). The successful formation of polymer is further confirmed by GPC characterization with a M_n of 4700 and PDI of 1.28 (Fig. 1C). From the terminal benzophenone group existence (~ 7.8 ppm in Fig. 1B), carbonyl group existence (verified by FT-IR and XPS spectra in Fig. 1D and Fig. 1E, respectively) and the appearance of C-OH group (verified by ^1H NMR, FT-IR and XPS spectra in Fig. 1B, Fig. 1D and Fig. 1E, respectively), the polymerization mechanism is verified as Barbier polyaddition. For a better understanding of Barbier reaction in polymerization, the polymerization mechanism is illustrated in Scheme 1 (This work), via both AB type and A₂+B₂ type Barbier polyaddition between Grignard reagent and carbonyl group in one pot. As the Grignard reagent forms, the addition of highly reactive Grignard reagent to the carbonyl group takes place successively in one pot. Because of the bifunctionality of the monomers, either AB type or A₂+B₂ type, continuous addition reactions between Grignard reagents and carbonyl groups happen, i.e., polyaddition mechanism, resulting in successful polymer synthesis. Other examples of chemical structures of monomers and the corresponding polymers synthesized via Barbier polyaddition are listed in Table 1. From the comparison of ^1H NMR spectra of these monomers and corresponding products (Supplementary Fig. S1-S8), the products clearly exhibit much broader signals for aromatic protons at $\sim 8.0\text{--}6.5$ ppm, hydroxide proton at $\sim 3.5\text{--}2.5$ ppm and alkyl protons at $\sim 2.5\text{--}1.0$ ppm, which is typical evidence of successful polymer synthesis.”.

We have revised our manuscript according to reviewer's comments. The point-by-point responses have been listed as follows:

1. All GPC traces in Fig. S2~S10 show a shoulder (or peak) at low molecular weight direction. In addition, the GPC curves should be in the range of 5~13 mL/min in order to display components of the formed polymers because the polymers obtained from step polymerization generally display multiple GPC curves except those polymers obtained via full precipitation separation, narrow and single GPC curves are unusual.

Response: Our GPC system is equipped with one TSK-Gel GMH_{HR}-N column, the signal after elution volume of 10 mL should be due to oligomers and then solvent. We therefore only reported results before 10 mL in the previous manuscript. According to reviewer's comment, we have provided all of GPC curves in the range of 5~13.5 mL for better illustration of the results. Because there are a lot of GPC results revised, we therefore do not provide the revised GPC results in this cover letter. Corresponding changes have been made and highlighted in the supporting information. Just as mentioned by the reviewer, the shoulder at lower molecular weight is a sign of step polymerization, even though, polymers exhibit narrow molecular weight distribution. We therefore revised the manuscript by deleting our previous statement of "unimodal and symmetric GPC curves", as follows: "It is worth mentioning that molecular weight distributions of the synthesized polymers are relatively narrow (PDI < 1.5) (Supplementary Fig. S9).".

2. "After stirring for 5 min, 0.1 mL of 1, 2-dibromoethane was added to the flask", the purpose for addition of 1, 2-dibromoethane into the reaction solution should be clearly explained.

Response: As is known, 1,2-dibromoethane can act as an activator in the Barbier reaction and Grignard reaction. Actually, 1,2-dibromoethane is not necessary in our research. However, the Grignard reagent formation is sluggish and affected by halide type and substituents. To reduce the initial reactivity differences between monomers containing different halides and substituents, we therefore chose 1,2-dibromoethane as the activating agent to initiate the sluggish Grignard reactions in the experiments. According to reviewer's comment, we have added the explanation in the experimental section, as follows: "0.1 mL of 1, 2-dibromoethane was added to the flask as activator".

3. Page 4, line 65-67, "It is worth mentioning that molecular weight distributions of the synthesized polymers are relatively narrow, with PDI smaller than 1.5, with a feature of chain-growth polymerization, rather than step-growth polymerization, which is shown by the unimodal and symmetric GPC curves". As discussed in the text, this is a polyaddition reaction of AB or A₂ + B₂ monomer, which determines that this is a step polymerization, not chain polymerization because in the polymerization, the reactions between the AB monomers, or oligomers with A and B groups respectively at their ends should occur. Obviously, the GPC curves shown in Fig.S2~Fig.S10 are artificially selected.

Response: Thanks for pointing out this issue, which is mentioned by Reviewer #2 as well. The reason that we raised this hypothesis of chain-growth polymerization was based on the narrow molecular weight distribution data. After careful thinking, we realize the raised hypothesis is wrong. The GPC curves obtained are NOT unimodal and symmetric. Shoulders can be seen. These results are typical feature of step polymerization. According to reviewer's comment, we have provided all of GPC curves in the range of 5~13.5 mL for better illustration of the results. To correct our statement, we have revised it as follows: "It is worth mentioning that molecular weight distributions of the synthesized polymers are relatively narrow (PDI < 1.5) (Supplementary Fig. S9).".

4. In the ¹H NMR spectra of Fig.2~Fig.9, the hydroxyl proton signal should be marked; and the signal(s) of phenyl proton(s) next to the carbonyl group seem to appear in the ¹H NMR spectra of the polymers, which refers to the monomers remained in the polymers, or to the terminal units. If the latter case is correct, the M_n can be calculated from the ¹H NMR data.

Response: Thanks for this great suggestion. In revised manuscript, the hydroxyl proton signals have been marked in the ^1H NMR spectra of Fig. 1 and Fig. S1~Fig. S8. The signals of phenyl protons next to the carbonyl group are the terminal units in the polymers. The M_n have been calculated from the ^1H NMR data by comparing the integral ratio between terminal group and polymer, which have been added in Table 1.

Table 1. The monomer structures and results of polymers synthesized via Barbier polyaddition in THF at 80 °C for 24 hour

No.	Monomer	Polymer	Abbreviation ^a	Yield(%)	$M_{n,NMR}(\text{Da})^b$	$M_n(\text{Da})^c$	PDI ^c	FL
1			PTPM-1	86.4	5600	5300	1.28	AIE
2			PTPM-1'	83.6	4600	4600	1.31	AIE
3			PTPM-2	76.6	5300	4800	1.25	ACQ
4			PDPM	61.8	5600	4300	1.21	Weak
5			PMPM	67.7	5400	5500	1.39	Weak
6			PXPM-1	55.7	6500	5800	1.47	Weak
7			PXPM-2	57.4	5600	4900	1.40	Weak
8			PXPM-3	57.0	5200	5600	1.45	Weak

^a triphenylmethanol polymer (PTPM), diphenylmethanol polymer (PDPM), monophenylmethanol polymer (PMPM) and xylylene-*alt*-phenylmethanol polymer (PXPM). ^b Calculated from ^1H NMR spectroscopy by comparing the integral ratio between terminal group and polymer. ^c Measured by GPC.

5. Note in Table 1 and S1 says that conversions were calculated based on the proton NMR data, based on the ^1H NMR spectra in Fig.S2~S9, this calculation may have big error because ^1H NMR spectra of the monomers are overlapped with that of their corresponding polymers. It is best to give the yields.

Response: We have removed conversion results from Table 1 and only provided the yields in revised manuscript, according to reviewer's comment.

6. It is better to merge Table 1 and Table S1 because Table S1 includes many data listed in Table 1.

Response: Thanks for this suggestion. In revised manuscript, we have replaced Table 1 with Table S1 in the manuscript and deleted Table S1 in the supporting information, according to reviewer's comment.

7. "Figure S15..... A) absorption spectrum, emission spectrum (excited @ 259 nm) of solution, B) excitation spectrum, emission spectrum (excited @ 274nm) of solution and digital photos (under sunlight and irradiation with UV lamp @ 365 nm)". "digital photos (under sunlight and irradiation with UV lamp @ 365 nm)" is in

Fig.S15A, not in Fig.15B, please check and revise!

Response: Sorry for making the mistake in the manuscript. We have checked and revised all of these mistakes in our manuscript, according to reviewer's comments. The corresponding changes in Figure S11-12 and Figure S20-22 have been highlight in the figure captions.

8. In Aggregation-induced emission (AIE) measurements section of the supporting information, "The fluorescence spectra and transmittance at 500 nm of polymers in THF/Water mixtures with different water content were recorded respectively on a Hitachi F2500 fluorescence spectrofluorometer and on a Shimadzu UV-2450 UV-Vis spectrophotometer", Water content in the THF/water mixture did not affect the size of polymer aggregates (all are 500 nm), this is not reasonable.

Response: We chose 500 nm as the wavelength of measuring transmittance, because PTPM-1 has no absorption at 500 nm (Figure D). Thus, the transmittance at 500 nm could reflect the turbidity of the solution more reasonably. The 500 nm here is wavelength, not particle size. To eliminate this misunderstanding, we have revised "500 nm" as "500 nm wavelength" in the main text and experimental section.

Figure D. The absorption spectrum of PTPM-1 solution in THF.

9. Page 5, line 89-90, "During the evaporation process of PTPM-1 solution, its photoluminescence is gradually induced". How was the conclusion "its photoluminescence is gradually induced" drawn based on Fig.1E? The phenomenon in Fig.1E should be clearly explained in order to get "its photoluminescence is gradually induced".

Response: As the solvent evaporates, the PTPM-1 solution spot becomes more and more concentrated. PTPM-1 will aggregate and get dried finally. Attributed to the AIE property of PTPM-1, this evaporation procedure will be accompanied with increasing photoluminescent intensity. According to reviewer's comment, we have revised the manuscript by adding explanation as follows: "During the evaporation process of PTPM-1 solution, PTPM-1 starts to aggregate from the solution. Attributed to the AIE property of PTPM-1, the PTPM-1 spot on the thin layer chromatography plate exhibits increasing photoluminescent intensity, as the solvent evaporates."

10. There is no number in scale bar in Fig.3G and 3H.

Response: Thanks for pointing out this issue. We have added the value of scale bars in Figure 3G and 3H (Figure 4G and 4H in the revised manuscript) in the caption, as follows: "Scale bar is 2.5 mm for all".

REVIEWERS' COMMENTS:

Reviewer #2 (Remarks to the Author):

The authors have adequately addressed my previous comments and suggestions. The revisions are satisfactory and the changes are acceptable. The quality of the manuscript has been improved after revision. I do not have further criticism of the work. The paper can now be accepted for publication as is.

Reviewer #3 (Remarks to the Author):

This manuscript has been carefully revised and the mistakes have been corrected, some necessary data have been supplemented according to the reviewer comments. However, some of the supplemented data (see the following comments) are not understandable, please check these data and revise the manuscript before this manuscript is accepted.

1. The manuscript shows many XPS O1s spectra, but did not explain what these spectra are used for. It is necessary to use of these spectra to explain the obtained polymer structure and further confirm the proposed polymerization mechanism. Generally, the peak areas can be used to estimate their relative content. However, in most of these Figures, the C=O peaks are too big, for example, the ratio of C=O/C-O is 1/29 based on the molecular weight of the PTPM-1, the areas ratio of C=O peak/C-O peak in Fig. 1E is much bigger than 1/29. How to explain this phenomenon?

2. After polymerization, the content of carbonyl group in the obtained polymers should be reduced significantly because there is only one carbonyl group in every polymer chain, but why did the carbonyl band in FT-IR spectra of Fig.S2, S3, S6, S7 and S8 is so strong.

We thank the reviewers for the constructive comments, which are very helpful for us to improve our manuscript. We have addressed each comment below and highlighted where the changes were made with yellow color. The point-to-point responses to the comments are attached.

REVIEWERS' COMMENTS:

Reviewer #2 (Remarks to the Author):

The authors have adequately addressed my previous comments and suggestions. The revisions are satisfactory and the changes are acceptable. The quality of the manuscript has been improved after revision. I do not have further criticism of the work. The paper can now be accepted for publication as is.

Response: Thanks for the suggestion of accepting our manuscript for publication and the efforts on our manuscript.

Reviewer #3 (Remarks to the Author):

This manuscript has been carefully revised and the mistakes have been corrected, some necessary data have been supplemented according to the reviewer comments. However, some of the supplemented data (see the following comments) are not understandable, please check these data and revise the manuscript before this manuscript is accepted.

Response: Thanks for the suggestion of accepting our manuscript for publication and the efforts on our manuscript.

1. The manuscript shows many XPS O1s spectra, but did not explain what these spectra are used for. It is necessary to use of these spectra to explain the obtained polymer structure and further confirm the proposed polymerization mechanism. Generally, the peak areas can be used to estimate their relative content. However, in most of these Figures, the C=O peaks are too big, for example, the ratio of C=O/C-O is 1/29 based on the molecular weight of the PTPM-1, the areas ratio of C=O peak/C-O peak in Fig. 1E is much bigger than 1/29. How to explain this phenomenon?

Response: Thanks for the suggestion on the explanation of XPS O1s spectra. Just as reviewer mentioned, the reason why we carried out XPS O1s tests in the previous revision, is to explain the obtained polymer structure and further confirm the proposed polymerization mechanism. The XPS O1s spectra with the coexistence of C=O and C-O bonds confirm the obtained polymer structure and the proposed polymerization mechanism. On the other hand, XPS is a surface-sensitive spectroscopic technique. The bigger area ratio of C=O peak/C-O peak should be due to the uneven distribution of terminal C=O group and repeating C-O groups in the polymer materials obtained after precipitation method. During the precipitation of polymer materials, the repeating C-O groups preferred to locate in the inner of polymer material, while the terminal C=O group preferred to locate on the surface, according to the XPS O1s spectra. According to reviewer's comments, we have revised the main text as follows: "To explain the obtained polymer structure and further confirm the proposed polymerization mechanism, FT-IR and XPS O1s characterizations were carried out. From the terminal benzophenone group existence (~7.8 ppm in

Fig. 2B), the carbonyl group existence (verified by FT-IR and XPS O1s spectra in Fig. 2D and Fig. 2E, respectively) and the appearance of C-OH group (verified by ^1H NMR, FT-IR and XPS O1s spectra in Fig. 2B, Fig. 2D and Fig. 2E, respectively), the polymer structure is verified to contain C-OH group and terminal benzophenone group, and the polymerization mechanism is verified as Barbier polyaddition. The relatively high ratio of C=O peak/C-O peak in XPS O1s spectra should be due to the uneven distribution of terminal C=O group and repeating C-O groups in the polymer materials obtained through precipitation method.”.

2. After polymerization, the content of carbonyl group in the obtained polymers should be reduced significantly because there is only one carbonyl group in every polymer chain, but why did the carbonyl band in FT-IR spectra of Fig.S2, S3, S6, S7 and S8 is so strong.

Response: FT-IR is not a suitable technique to quantitatively compare the content of different chemical bonds, because different chemical bonds have different FT-IR absorption capabilities. Carbonyl bond exhibits relatively strong absorption in FT-IR test, which causes the strong absorption signal, even though it is just terminal group in polymer chain. But, FT-IR is very suitable to determine the existence/inexistence of specific chemical bonds. For example, it is used to determine the existence of C=O bond and C-O bond in this manuscript, which is used to explain the obtained polymer structure and further confirm the proposed polymerization mechanism.